# Tissue Reactions and Mechanism in Cardiovascular Diseases Induced by Radiation

**DOI:** 10.3390/ijms232314786

**Published:** 2022-11-26

**Authors:** Xiao-Chang Liu, Ping-Kun Zhou

**Affiliations:** Beijing Key Laboratory for Radiobiology, Department of Radiation Biology, Beijing Institute of Radiation Medicine, Beijing 100850, China

**Keywords:** radiation, atherosclerosis, cardiovascular diseases

## Abstract

The long-term survival rate of cancer patients has been increasing as a result of advances in treatments and precise medical management. The evidence has accumulated that the incidence and mortality of non-cancer diseases have increased along with the increase in survival time and long-term survival rate of cancer patients after radiotherapy. The risk of cardiovascular disease as a radiation late effect of tissue damage reactions is becoming a critical challenge and attracts great concern. Epidemiological research and clinical trials have clearly shown the close association between the development of cardiovascular disease in long-term cancer survivors and radiation exposure. Experimental biological data also strongly supports the above statement. Cardiovascular diseases can occur decades post-irradiation, and from initiation and development to illness, there is a complicated process, including direct and indirect damage of endothelial cells by radiation, acute vasculitis with neutrophil invasion, endothelial dysfunction, altered permeability, tissue reactions, capillary-like network loss, and activation of coagulator mechanisms, fibrosis, and atherosclerosis. We summarize the most recent literature on the tissue reactions and mechanisms that contribute to the development of radiation-induced cardiovascular diseases (RICVD) and provide biological knowledge for building preventative strategies.

## 1. Introduction

The long-term survival rate of cancer patients has been rising in recent years due to the innovation and improvement of treatments and precise medical management. Radiation is a major approach to cancer treatment, and more than 50% of cancer patients have received radiotherapy [1,2]. Consequently, the risk of non-cancer diseases as the late effect consequences of tissue damage reactions is becoming a critical challenge. Evidence has accumulated that the incidence and mortality of non-cancer diseases have also increased along with the increase in survival time and long-term survival rate of cancer patients after radiotherapy [3,4]. Cardiovascular disease is a class of common non-cancer diseases and has drawn wide attention and concern [5,6]. According to epidemiological research and clinical trials [7,8], the development of cardiovascular disease in long-term survivors of chest cancer who received radiotherapy has been directly linked to chest radiation exposures, in which vascular injuries and progressive pathogenic alterations play a very important role. It has been established that people who undergo radiotherapy for malignant tumors are more prone to acquiring cardiovascular disease as their life expectancy grows. Even a low dose of occupational radiation exposure can increase non-cancer disease risks, such as cardio- and cerebro-vascular disease [9,10].

Radiotherapy can cause direct and indirect damage of endothelial cells by radiation, acute vasculitis with neutrophil invasion, endothelial dysfunction, altered permeability, tissue reactions, capillary-like network loss, and activation of coagulator mechanisms, fibrosis, and atherosclerosis, ultimately leading to a range of cardiovascular disorders. RICVD include coronary/carotid/peripheral artery diseases, cardiomyopathy, valve disease, pericardial disease; conduction abnormalities and arrhythmias, and hemodynamic problems [11,12,13,14]. Despite advances in radiation therapy technologies and cardiovascular shielding methods, RICVD has become a major cause of death for long-term survival cancer patients [15,16,17,18]. As a result, enhancing our understanding of cardiovascular disorders induced by radiation is critical for improving patients’ quality of life and extending their survival time. We summarize the most recent literature on the pathogenic pathways that contribute to the development of RICVD and provide preventative and therapeutic ideas.

## 2. Radiation-Induced Injuries of Cardiovascular Tissues

Nuclear genomic DNA is the critical target of ionizing radiation (IR), and DNA damage is ascertained to be a key molecular event leading to the cytotoxicity of radiation. Mammalian cells have evolved multiple DNA damage response signaling pathways [19] and diverse machineries for repairing every type of spontaneously occurring as well as exogenous factors-induced DNA damage [20]. DNA double-strand break (DSB) is the most severe form of DNA damage induced by ionizing radiation. There are at least three pathways for repairing DSBs, non-homologous end joining (NHEJ), homologous recombination (HR), and alternative end joining (alt-EJ). The dysfunction of cellular intrinsic DNA damage response and repair machineries can cause cells to become more sensitive to radiation. Radiation can affect various compartments of cardiovascular tissues, such as the coronary arteries, heart valves, pericardium, myocardium, and cardiac conduction system [21]. Vascular endothelial cell dysfunction is a leading cause of RICVD. It is currently thought that initial endothelial injury is induced by IR and that subsequent hemodynamic disturbances cause inflammatory reactions. In addition, IR can also change lipid metabolism and causes hyperlipidemia, which is one of the most important factors leading to atherosclerotic plaque [22]. Coronary-artery and cerebrovascular diseases are late manifestations of atherosclerotic changes of the arteries, representing the principal causes of vascular disease mortality and morbidity (Figure 1).

### 2.1. Early Effects of Radiation on Vascular Cells

Early tissue reactions usually occur from hours to a few weeks after IR, which may be of an inflammatory nature due to the changes in cell permeability, the release of inflammatory mediators, etc. Subsequent reactions are often a consequence of cell depletion, although non-cytotoxic effects on tissues also contribute to the early reactions. The manifestations of tissue injury induced by a given dose of radiation vary from one tissue to another depending on cellular composition, proliferation rate, and mechanistic signaling of response to IR. Endothelial cell damage attributes to multiple tissue injuries. For example, gastrointestinal reactions have been attributed to the radiation damage response of microvascular endothelial cells [23]. The villous epithelium cannot recover due to endothelial dysfunction and elevated inflammatory mediators, which results in the loss of the epithelial barrier [24]. Vessel endothelial injury can occur either in acute or early stage, or late stage, after IR, which is dependent not only on radiation dose received and the induction of cell killing, but also on the interactions of cell compositions, immune and inflammatory reactions, and metabolic changes.

The endothelium is one of the most vulnerable components of the vascular wall. Human umbilical vein endothelial cells (HUVECs) have been mostly used to study the effects and mechanisms of radiation on blood vessels at the cellular level in vitro. In addition to the canonical nuclear genomic DNA damage response and proliferation inhibition [20,25,26], other various forms of radiation effects on HUVECs were observed, which include senescence-associated secretory phenotype (SASP) and premature senescence [27,28,29], increased intrinsic ROS and ${\mathrm{NO}}_{2}^{-}$ production and release of pro-inflammatory cytokines [30,31,32,33], decreasing mitochondrial membrane potential [31], suppression of the autophagic flux [34], endothelial-to-mesenchymal transition (EndMT) [35], decreasing the capacity of forming capillary-like network [36,37], and increased permeability and disruption of cellular junctions [33]. The types of cell death of HUVECs induced by radiation include apoptosis [38], pyroptosis [39], and ferroptosis [40]. In addition to the direct targeting effects of radiation on HUVECs, significant radiation-mimic effects can also be evoked by the extracellular vesicles or exosomes, secreted from other types of tissue cells with or without IR, such as mesenchymal stem cells [41] and cancer cells [42,43,44]. This form of effect is called the radiation bystander effect [43,45], and it may have important practical implications in clinical cancer radiotherapy.

Endothelin-1 is a strong vasoconstrictor peptide thought to be causally implicated in hypertension and the development of cardiovascular disease. Endothelin-1 levels rise in in vitro cultured HUVECs and in vivo after radiation exposure at doses from 0.2 Gy low dose to 20 Gy high dose [46,47,48,49]. Furthermore, angiotensin II synthesis and release by bovine pulmonary artery endothelial cells and HUVECs, as well as endothelial cells from irradiated rats’ lung endothelial cells, demonstrated a dose- and time-dependent increase beginning 24 h after 5–30 Gy IR [50,51,52,53]. This implies that the endothelium causes initial vasodilatation following IR, followed by persistent vasoconstriction and that endothelium-dependent vasodilation is hindered.

Vascular smooth muscle cells (VSMCs) are another fundamental component of blood vessels and are essential for arterial physiology and pathology. Apoptotic VSMCs can expedite the instability and inflammation of atherosclerotic lesions. IR can directly impact VSMCs in addition to the vascular endothelium. IR exposure from a dose of 1.25 Gy can reduce VSMC proliferation [54,55,56]. IR induces apoptosis of VSMCs associated with the increase in C-reactive protein [57].

### 2.2. Late Effects of Radiation on Cardiovascular Tissues

Tissues of the human body vary widely in the proliferating rates at which their component cells. The rapidly proliferating tissues, such as bone marrow and epithelium of the intestinal mucosa, have a population of stem cells that give rise to a proliferating cell compartment. Some other types of tissues possess large populations of functional mature cells that may have very limited division capability to help in restoring function upon some cell depletion. In general, the timing of radiation-induced tissue injury depends on the life span of the mature cells. Tissue damage after irradiation manifests earlier for those rapidly proliferating tissues with radiosensitive stem cells. Whereas the tissue damage in response to radiation is comparatively delayed for those slow turnover tissues with relatively radioresistant mature cells. For the blood vessels where endothelial cell turnover is very low, the timing of the response of such tissues to radiation is dose-dependent but may not be evident until a long time after IR. Circulatory disease has been recognized as a major late effect of IR exposure, both for mortality and morbidity.

Radiation therapy evokes endothelial inflammation, which exacerbates intimal vessel damage, and finally causes atherosclerotic plaques to develop in the coronary arteries, leading to cardiovascular diseases. After radiation, coronary heart disease might manifest itself 5 to 20 years later [58,59].

A cohort of roughly 86,000 atomic bomb survivors who received 0 to 3 Gy of IR was monitored. Heart disease killed around 8400, or roughly 10%, of the observation group [60]. The predicted risk of heart disease is 14% per Gy of IR [60]. According to experimental research, IR damage to the capillary network is a potential cause of myocardial degeneration and heart failure following radiation exposure. Clinical trials in asymptomatic breast cancer patients have also shown regional perfusion abnormalities six months to five years after IR. The greatest of these investigations revealed that the incidence of perfusion defects was clearly related to the amount of left ventricular volume included in the irradiated field: perfusion was reduced by 10 to 20%, whereas volume perfusion was reduced by 50 to 60%, 5%, and 0.5%, respectively [61].

Technetium-99m methoxyisobutylisonitril SPECT myocardial perfusion imaging revealed that 27% of patients had myocardial ischemia at 6 months after radiotherapy, 42% had myocardial ischemia at 24 months after radiotherapy, and 27% had myocardial ischemia at 6 months after radiotherapy, in a study of 114 patients with asymptomatic breast cancer who had received radiotherapy. Furthermore, the ischemia region is linked to the radiotherapy site and dosage [62]. Other epidemiological studies have clearly shown that radiotherapy is an independent risk factor for 10-year cardiovascular death after thoracic radiotherapy of Hodgkin’s lymphoma and breast cancer [7,12,61,63,64,65,66,67], as well as cerebrovascular death after head and neck cancer radiotherapy or craniocerebral radiotherapy in children [68].

## 3. Risks of Cardiovascular Diseases in Different Scenarios of Radiation Exposures

Although the cardiovascular system, particularly the heart, was formerly assumed to be rather resistant to radiation-induced damage [69], RICVD is a recognized source of morbidity and death in radiotherapy patients [3,4,5,70], and low-dose, or occupational, radiation exposures [71,72,73]. Coronary artery disease (CAD), valvular heart disease, pericardial disease, heart failure (HF), right ventricular injury, arrhythmia, peripheral arterial disease, systemic hypertension, pulmonary hypertension, and thromboembolism disease are currently established as radiation-related cardiovascular complications [4,74,75,76].

### 3.1. Risk of Cardiovascular Diseases Related to Radiotherapy

Radiotherapy-related cardiovascular problems were initially identified in the 1970s. A study of 46 individuals treated with chest radiation discovered that radiotherapy produced heart fibrosis (involving the endocardium, myocardium, and pericardium), and pericarditis was the most common clinical manifestation [77]. Radiation-induced arterial stenosis has been related to a variety of pathogenic processes, including damage and blockage of blood vessels supplying the arterial wall [78,79] and endothelial dysfunction [78,80,81,82]. Indeed, prior clinical research has demonstrated that IR is a separate risk factor for the development of early atherosclerosis [83,84]. Cancer survivors who received mediastinal irradiation have a 1.5 to 3 times greater risk of fatal myocardial infarction than cancer survivors who do not receive radiotherapy [85,86]. Cardiomyopathy, defined as a leading cause of reduced left ventricular ejection fraction (LVEF) and associated with high mortality risk from progressive heart failure and arrhythmias, was found to be more prevalent in radiotherapy-exposed survivors than in non-radiotherapy-exposed patients in a study (odds ratio 1.9, 95% confidence range 1.1 to 3.7) [87]. In another study, individuals who received more than 3.7 Gy of radiation were found to have an 18% chance of developing heart failure 20 years later [87].

Angina and myocardial infarction were more common in patients with left-sided breast cancer treated with radiotherapy than in those with right-sided breast cancer, according to a study based on observations of 34,825 breast cancer patients with radiotherapy in Sweden and Denmark. Furthermore, radiation exposure to the heart averaged 6.3 Gy on the left side and 2.7 Gy on the right side; thus, a link was clearly demonstrated between the radiation dosage administered to the heart and CAD occurrences [88]. After undergoing mediastinal irradiation with a radiation dosage ranging from 25 to 42 Gy, CAD was discovered in 10.4% of patients with at least 20 years of follow-up [89]. Furthermore, the most common causes of mortality among Hodgkin’s lymphoma patients treated with radiation were primary or secondary malignancies and CVD [90,91]. The major conclusion of research to measure the risk of heart disease following radiotherapy based on radiation dosage is that a greater radiation dose to all individual cardiac components except the right atrium is related to an increased risk of heart disease occurrences. Although the percentage increase in risk caused by radiation is comparable in patients with and without coronary artery calcification (CAC), the absolute increase in risk is significantly larger in people with CAC [92].

### 3.2. Risk of Cardiovascular Diseases by Low-Dose Radiation Exposure

In vitro endothelial cell cultures treated with 0.125–0.5 Gy irradiation showed increased ICAM-1 expression and leukocyte attachment [93]. In addition, increased levels of IL-6 and CCL2 have been seen in human endothelium cells exposed to 0.5 Gy [94]. Epidemiological studies of Chornobyl disaster survivors have revealed that radiation exposures as low as 0.05–1.0 Gy may increase the risk of cardiovascular disease [95,96,97]. Evidence from the atomic bomb survivors’ life span study (LSS) also supports an elevated risk of myocardial infarction and stroke at very low dose levels, below 5 Gy and at mean doses just below 0.5 Gy [98,99]. The increased risk presented as the stroke mortality (excess relative risks (ERR)/Gy = 0.09; 95% CI: 0.01; 0.17) and heart disease mortality (ERR/Gy = 0.14; 95% CI: 0.06; 0.23), between 1950 and 2003.

Little et al. undertook a more in-depth investigation of the meta-analysis [100]. The authors chose studies published after 1990, and the exposure technique was whole-body IR with a cumulative mean dose of less than 0.5 Sv or a dose rate of less than 10 mSv/day). Four categories of circulatory system disorders, ischemic heart disease (IHD), non-IHD, cerebrovascular diseases (CVA), and other circulatory system diseases, were assessed for their excess risks. The results of meta-analysis using a random-effects model displayed a statistically significant ERR per Sievert for IHD (ERR = 0.1/Sv, 95% CI: 0.04, 0.15, 1-side *p* < 0.001), CVA (ERR = 0.21/Sv, 95% CI: 0.02, 0.39, 1-side *p* = 0.014), and circulatory disease apart from heart disease and stroke (ERR = 0.19/Sv, 95% CI: –0.00, 0.38, 1-sided *p* = 0.026; –0.00 indicates that the number is between –0.005 and 0). The population excess risk estimates for all cardiovascular disorders ranged from 2.5% per Sv (France, 95% CI, 0.8 to 4.2) to 8.5% per Sv (Russia, 95% CI, 4.0 to 13.0).

The tissues and organs of interest frequently get doses of well over 0.5 Gy during radiation treatment, which is generally consistent with the population risk at the low dose/dose rates indicated above. This shows that the processes at large doses and high dose rates may be comparable to those at low doses and dose rates. The fact that IHD risks in high-dose/partial-body exposed groups are similar to those in typically whole-body exposed groups in response to whole-body dosage suggests that mean heart dose may be the most important measure for predicting radiation-associated IHD [101].

The connection between childhood exposure to low to moderate levels of IR to the head and neck and the development of vascular illness (IHD, carotid artery stenosis (CAS), and stroke) in adulthood was investigated in a cohort study of 17,734 Israelis. The average dosages to the brain, thyroid, salivary gland and breast were 1.5, 0.09, 0.78, and 0.017Gy, respectively. IR exposure increased the chance of getting any vascular disease (RR = 1.19, 95%CI: 1.09, 1.29), stroke (RR = 1.35, 1.20, 1.53), CAS (RR = 1.32, 1.06, 1.64), and IHD (RR = 1.13, 1.01, 1.26), after controlling for age, gender, socioeconomic status, smoking, hypertension, and diabetes [102].

There is a recent report on the association between low-dose external occupational radiation exposure and circulatory disease morbidity among diagnostic medical radiation workers [10]. In this report, ERR/100 mGy was estimated based on the analysis of a cohort of 11,500 diagnostic medical radiation workers in South Korea. ERR/100 mGy for all circulatory diseases was 0.14 (95% CI −0.57 to 0.99). There is a non-significant increase in the radiation risks of cerebrovascular diseases and ischemic heart diseases, with estimates of individual cumulative doses to the heart [ERR/100 mGy = 3.10 (−0.75 to 11.59) and 1.22 (−0.71 to 4.73), respectively]. ERR estimates were generally more strongly positive for female versus male workers and younger workers versus workers over 50 years old.

## 4. Mechanisms of Radiation-Induced Cardiovascular Injuries

The pathogenic processing and mechanisms of cardiovascular diseases caused by IR have not been fully studied, although it is certain that it is, at least partially, causing or evoking atherosclerosis formation. Atherosclerosis is a complex disease caused by genetic and environmental interactions that include the influence of IR. The endothelium is a fundamental structure and function layer of vascular, and its dysfunction plays an important role in the development of cardiovascular diseases. Radiation results in endothelial dysfunction on multi-aspects: targeting DNA damage, oxidative stress, inflammatory reaction, aging, cell death, permeability, etc.

### 4.1. ROS and Endothelial Dysfunction

Endothelial cells form a single layer of cells that line the inside of the vascular system, serving as a bridge between the blood and the surrounding tissues. Endothelial cells are engaged in a variety of physiological activities, including the regulation of vascular tone and permeability, which are necessary for optimal vascular function [103]. Endothelial dysfunction is seen in individuals with atherosclerosis as well as those exposed to CVD risk factors such as smoking, dyslipidemia, obesity, and diabetes [104], and it is regarded as one of the early markers of cardiovascular morbidity and death [105,106,107,108]. As a result, endothelial cells are thought to be a significant target for radiation-induced CVD.

Oxidative stress is a typical underlying biological process that leads to endothelial dysfunction [109]. An imbalance between the formation of ROS and the activity of enzymatic and non-enzymatic antioxidant mechanisms is characterized as oxidative stress [110]. Higher levels of ROS injure cells by causing oxidative damage to DNA, lipids, and proteins, which can lead to cell death [111]. There are comprehensive reviews regarding the involvement of oxidative stress in radiation-induced endothelial dysfunction [112] and cardiovascular toxicities [113,114].

### 4.2. Mitochondrial Dysfunction

Radiation-induced mitochondrial dysfunction is well-documented with regard to its influential role in CVD [115]. Mitochondrial dysfunction is linked to oxidative stress because mitochondria are both a target and producer of ROS. Oxidative stress can be noticed following irradiation for extended lengths of time due to increased ROS generation in the mitochondrial pathway [31,116]. Although other sources may contribute, mitochondria are assumed to be the primary source of these radiation-induced secondary ROS. For example, Leach et al. demonstrated that, between 1 and 10 Gy, the amount of ROS-producing cells increased with the dose, which they suggested was dependent on the radiation-induced propagation of mitochondrial permeability transition via a Ca^2+^-dependent mechanism [117].

At the same time, mitochondria, particularly mitochondrial DNA (mtDNA), are important targets for ROS. Various studies have found an increase in the accumulated common deletions (CDs) of mtDNA following IR exposure [118,119,120]. Following exposure to doses as low as 0.1 Gy, accumulated CDs can be detected in numerous human fibroblast cell lines. Surprisingly, higher CD accumulation was found in bystander cells, which were evoked by the conditioned media generated from 0.1 Gy irradiated cells. Mitochondria are also important apoptosis executors and playmakers of senescence [121] and inflammation [122]. IR induces mitochondrial respiratory chain complex dysfunction and superoxide generation [123,124], which results in long-term senescence of endothelial cells [124]. Dysregulation of all these mitochondria-associated processes is closely linked to the formation and progression of atherosclerosis [125,126,127,128].

### 4.3. Endothelial Senescence and Inflammation in Atherosclerotic Plaque Formation

The initial events of radiation-induced atherosclerosis are endothelial cell damage and the migration of monocytes into the intima, followed by the uptake of LDL and the formation of distinct streaks. Radiation-induced DNA damage, especially DSB, is a major cause of cell death, senescence, and the dysregulation of proliferation and differentiation [129]. Furthermore, oxidative stress is a substantial contributor to radiation-induced senescence, as well as radiation-induced DNA damage and faster telomere attrition [129,130,131,132,133]. IR has been proven in multiple in vitro investigations to promote endothelial cell senescence, primarily by increasing doses of radiation [134,135]. The majority of studies agree that radiation-induced premature endothelium senescence is caused by activating canonical DNA damage response, which is comparable to replication senescence [136]. Senescent endothelium cells further produce pro-inflammatory cytokines such as interleukin-1, IL-6, IL-8, and transforming growth factor (TGF-β), as well as ROS (superoxide, peroxynitrite) [124,137,138] (Figure 2). All these molecules, as indirectly evidenced by the presence of senescent endothelial cells in human atherosclerotic plaques, contribute to the formation and progression of atherosclerosis [139,140]. These reactions cause monocyte recruitment and adherence, as well as migration to the endothelium, where they change into activated macrophages in the context of high cholesterol levels. The activated macrophages then take up lipids, transform into foam cells, and produce fatty streaks in the intima, leading to atherosclerotic plaque development and hastening myocardial infarction [141].

For example, Kim et al. discovered that 4 Gy exposure resulted in a senescence phenotype in endothelial cells, shown mechanically as increased expression of p53 and p21 and downregulated cyclin and Rb phosphorylation [142]. The premature senescence of endothelial cells was demonstrated after being subjected to chronic low-dose-rate radiation (1.4, 2.4, and 4.1 mGy/h) for 1, 3, and 6 weeks [143]. A dosage rate threshold was established for the onset of premature senescence. The beginning of senescence was not accelerated by 1.4 mGy/h exposure, while it was accelerated by 2.4 and 4.1 mGy/h exposure. Notably, when the total dosage absorbed by the cells reached 4 Gy, a senescence curve was detected. Proteomic research demonstrated the importance of radiation-induced oxidative stress and DNA damage in activating the p53/p21 pathway. The PI3K/Akt/mTOR pathway has also been postulated to play a role [144].

Endothelial inflammation is an important risk factor in the initiation and development of cardiovascular disease. IR causes vascular inflammation, including acute vasculitis with neutrophil invasion and the prolonged inflammation associated with senescence [145]. TGF-β, a pro-inflammatory factor released by vascular endothelial cells after IR, paracrine-ly activates the classical Smad-dependent signal pathway in cardiomyocytes and promotes cardiac muscle fibrosis. Furthermore, by phosphorylating TGF-activated kinase 1, such as phosphorylating extracellular signal-regulated kinase and p38 mitogen-activated protein kinase (MAPK), TGF-β can initiate a series of mitogen-activated protein kinase cascades. Activation of P38 (MAPK) and Jun kinase activate the non-Smad-dependent signaling pathway [146,147], which causes cardiomyocyte apoptosis, fibroblast proliferation, and collagen deposition, as well as accelerates the process of myocardial fibrosis.

IR-induced endothelial inflammatory reactions release a large number of cytokines and growth factors. In addition to evoking endothelial dysfunction, they work together to promote the stillness of vascular smooth muscle cells to differentiate into an active state of synthesis. They are more capable of proliferation, migration, and secretion in this state. Smooth muscle cells migrate to the endometrium, proliferate, and increase the production of extracellular matrix [148]. This pathological process that results in intimal thickening paves the way for the formation of atherosclerotic plaques.

### 4.4. Late Tissue Reactions to Cardiovascular Diseases

#### 4.4.1. Vascular Stenosis

Vascular stenosis is a common form of radiation late-effect, which forms even decades after IR exposure [149]. After the atherosclerotic process begins, lipid-loaded foam cells emit inflammatory cytokines and growth factors, encouraging smooth muscle cell proliferation and migration. Atherosclerosis then either advances to a stable advanced lesion with a thick protective fibrous cap or grows in size if a fibrous cap is not present. These initially huge lesions are loaded with inflammatory cells and can easily burst, resulting in a deadly myocardial infarction or stroke; or else, the ruptured lesion may gradually mend but leave a small artery with a substantially diminished lumen [150]. A follow-up study on neck radiotherapy patients indicated that there exists long-term radiation-induced carotid artery stenosis, demonstrating pathologically as intima-media fibrosis, endothelial cell loss, and decreased expression of thrombomodulin [149].

#### 4.4.2. Atherosclerosis

The intima is made up of an endothelial-cells layer situated on a thin basement membrane that runs between subendothelial connective tissue and the smooth muscle cells that lie underneath it. Smooth muscle cells and elastic networks make up the tunica media. The adventitia is a hazy layer of connective tissue that is interlaced with elastic and nerve fibers as well as tiny trophic blood vessels. The veins, which bring deoxygenated blood back to the heart, lack the three distinct adventitia seen in the arteries. Capillaries are made up of a single layer of endothelial cells with no underlying smooth muscle cells.

Radiation-induced alterations in the microcirculation promote inflammation, which can lead to microthrombi and tissue ischemia when combined with endothelial cell detachment and subendothelial cell exposure. Monocytes attach to irradiated endothelial cells and move into the intima of bigger arteries. In the presence of elevated cholesterol, invading monocytes convert into activated macrophages (foam cells), which eat lipids and generate fatty streaks in the intima, therefore commencing the atherosclerosis process. Ischemia produced by radiation-induced microvascular damage and vessel stenosis may result in organ atrophy and fibrosis of the myocardium.

Atherosclerotic plaques and plaque rupture with thrombosis are significant causes of the acute coronary syndrome, cerebrovascular accidents, and peripheral ischemia episodes in the large and medium arteries. Notably, coronary microvascular disease and dysfunction are becoming more recognized as risk factors for heart failure with intact ejection fraction.

#### 4.4.3. Fibrosis

Myocardial and vascular fibrosis is another form of late tissue reaction to cardiovascular diseases [151]. Inflammatory substances and other cytokines, such as thrombomodulin and plasminogen activator inhibitor (PAI)-1, damage endothelial cells and promote thrombosis and atherosclerosis, resulting in profibrosis [152,153]. Increased levels of TGF-β, for example, cause changes in the extracellular matrix.

The TGF-β-1 signaling axis is the canonical pathway of fibrosis development, in which TGF-β-1 first binds to heterodimeric receptors in the plasma membrane composed of TGF-β-type I and TGF-β-type II hemi-receptors, and then induces phosphorylation of Smad2 and Smad3 transcription factors in the cytoplasm that are complex with Smad4. Transcription factors then move to the nucleus and form transcription complexes that transactivate important ECM genes such as type I collagen [154]. Connective tissue growth factor (CTGF), a protein found in stromal cells, is hypothesized to be a downstream mediator of TGF-β-1-induced fibroblast activation. Furthermore, CTGF might be activated independently of TGF-β-1 and Smad signaling by activating the Rho/ROCK pathway [155]. TGF-β receptor activation can also generate nonclassical signals that activate the p42/p44 MAPK cascade in fibroblasts, driving myofibroblast transformation via signals involving the signaling effector calcineurin and many transcription factors, in addition to Smad-mediated pathways. Transformation and differentiation of cardiac fibroblasts into contractile and secretory myofibroblasts is a key cellular event that drives the healing and fibrosis processes following radiation exposure. [156,157,158,159].

#### 4.4.4. Role of Tissue Level Bystander Effects in Cardiovascular Diseases

The radiation-induced bystander effect (RIBE) or radiation-non-targeted effect at the tissue level refers to a wide range of biological phenomena that occur in unirradiated cells or tissue/organs as a result of radiation-mimic signal transmission from irradiated cells of normal tissues/organs or tumors [160]. The bystander cells behave similarly to directly irradiated cells, exhibiting increased genomic DNA damage, apoptosis or cell death, increased mutagenesis, decreased clonogenic efficiency, senesces and inflammation, etc.

Macrophages are thought to be important players in RIBE. One theory is that the macrophage response is caused by phagocytosis by apoptotic cells rather than direct irradiation [161]. This may cause macrophages to release tumor necrosis factor (TNF)-α, which has the ability to cause DNA damage [162]. ROS signaling could be another mechanism linking macrophages and inflammation to RIBE.

The exosomes generated from the irradiation tissues or tumors are believed to be major mediators of triggering whole-body bystander effects [163,164,165,166]. Increasing evidence shows that the exosome plays an important role in cardiovascular diseases [167,168,169]. Exosomes are extracellular vesicles ~40 to 160 nm in diameter with an endosomal origin, which contain many constituents of a cell, including DNA, RNA, lipids, metabolites, and cytosolic and cell-surface proteins. The exosomes are speculated to have the role of removing excess and/or unnecessary constituents from cells to maintain cellular homeostasis, mediating intercellular communication and the systemic abscopal effect (in radiotherapy). Multi-faceted applications are assumed for exosomes, such as diagnostic markers, prognostic markers, involvement in pathological processes, and therapeutic agents in cardiovascular diseases, which largely depend on the source and inclusions of the exosomes. However, the mechanism and risk weight factor of exosomes in radiation exposure-related cardiovascular diseases need more in-depth and extensive research.

## 5. Prospective and Conclusions

Cancer patients have traditionally been excluded from all major cardiovascular clinical trials and large prospective studies. Recent advances in cancer treatment and improved patient survival have raised awareness of cardiovascular toxicity and premature cardiovascular disease in cancer survivors, necessitating more comprehensive studies in such patient populations. RICVD is one of the most common types of cardiovascular disease in cancer survivors, and its underlying pathogenic process and characteristics, and risk levels are not yet fully understood.

At the moment, the most commonly used radiation-damage treatments are: preventive and curative medications (drugs that can reduce radiation damage, also known as anti-radiation drugs), anti-infection (antibiotic drugs should be used according to the type of infection), anti-hemorrhage (platelet replenishment and protection, maintenance and improvement of vascular function, and correction of blood clotting disorders), blood transfusion (fresh (frozen) plasma and its organic fraction), nutritional therapy and substance metabolic balance (changes in water, electrolytes, and acid-base balance should be checked and corrected in a timely manner), hematopoietic factors (CSFs, EPO, TPO, etc.), stem cell transplantation (HSCT), and other symptomatic treatment and psychological care (to strengthen psychological care, understand the patient’s psychological status, provide timely and focused counseling and explanation, reduce the patient’s psychological burden, and enhance confidence in overcoming the disease).

There are numerous avenues for more robust research and development. Continued advances in RT technology may reduce the dose of radiation to the heart and its substructures even further. The use of machine learning algorithms to analyze current imaging datasets may aid in detection analysis. Short- and long-term patient follow-up following radiation therapy administration can be used to detect and adjudicate the occurrence of cardiovascular events and identify predictors. Furthermore, the early involvement of cardiac oncologists in the care of patients undergoing thoracic RT may speed up the detection, intervention, and optimization of pre-existing cardiac risks. On the biological aspect, advances in the genomic analysis may aid in identifying patients at increased risk of radiation-induced heart disease. Similarly, more research into cardiovascular toxicity biomarkers may lead to earlier detection and treatment. The development of advanced treatments and options may accelerate the recovery and healing of damaged cardiac tissue for patients who already have significant cardiac dysfunction. Given the rapidly evolving field of cardiovascular research, the future of RICVD research will continue to necessitate collaborations between radiation oncologists, cardiologists, and basic scientists, as well as the application of a team science approach.

In conclusion, RICVD remains a leading side-effect and cause of morbidity and mortality among cancer survivors, despite recent advances in radiation therapy. Prevention strategies such as new radiation management techniques, minimizing cardiovascular radiation exposure, lowering radiation dose, treating cardiovascular risk factors, and possibly certain medications for prevention in high-risk groups may all help to lower the risk of heart disease. Although large-scale data are lacking, the clinical judgment suggests that active management of traditional risk factors such as hypertension, diabetes, hyperlipidemia, smoking, and smoking cessation is the best management for patients receiving MRT or radiation involving the cardiac area. With the unique characteristics and needs of the growing RICVD patient population, disease-specific basic and clinical research, randomized clinical trials, and prospective studies are required to elucidate the disease’s exact pathogenesis and to establish safe and effective treatment options.

## Figures and Tables

**Figure 1 ijms-23-14786-f001:**
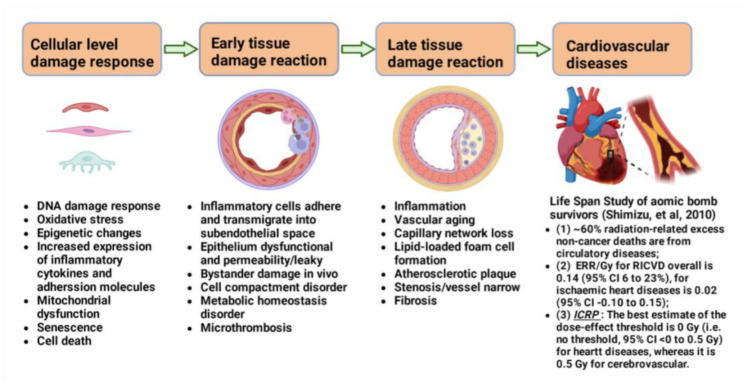
Development of RICVD. Radiotherapy can cause direct and indirect damage of endothelial cells by radiation, acute vasculitis with neutrophil invasion, endothelial dysfunction, altered permeability, tissue reactions, capillary-like network loss, and activation of coagulator mechanisms, fibrosis, and atherosclerosis, ultimately leading to a range of cardiovascular disorders. The main factors at the cellular level are reactive oxygen species (ROS), mitochondrial dysfunction, endothelial dysfunction, and inflammation. Vascular stenosis, atherosclerosis, and fibrosis are examples of late tissue responses in cardiovascular disease.

**Figure 2 ijms-23-14786-f002:**
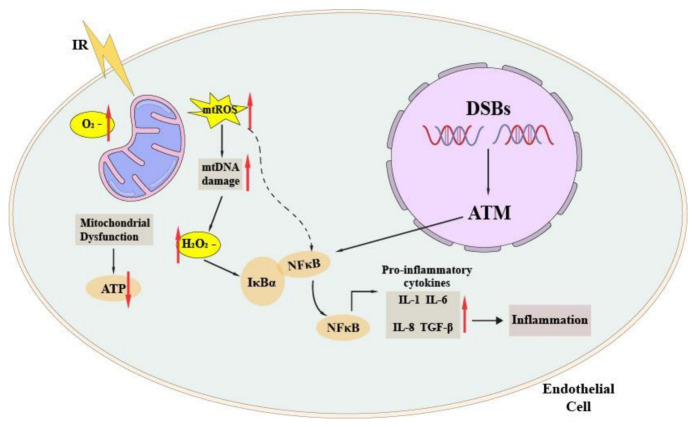
Radiation-induced inflammation in endothelial cells. IR exposure activation of redox-sensitive transcription factor NF-κB via DSB and oxidative stress. The resulting inflammation leads to the production and secretion of pro-inflammatory cytokines.

## Data Availability

Not applicable.

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
