# Peer review of "Tissue Reactions and Mechanism in Cardiovascular Diseases Induced by Radiation"

_ijms, 2022, doi:10.3390/ijms232314786_

Round 1
Reviewer 1 Report
This manuscript addresses a topic of great interest to biomedical researchers, especially to those who are studying radiobiology and the diagnostic and therapeutic applications of ionizing radiation.
However, in its current form, the review is organized in a very confused manner, with the same concepts being repeated over and over. Among extensively repeated topics are inflammation of the endothelium caused by ionizing radiation, the role of ROS in the induction of vascular damage, and the atherogenesis process. It would be much better to group information relating to a specific topic in a specific paragraph, in which all the introductory information will be provided, avoiding repeating them several times.
Furthermore, the text is full of vague statements. See, for example, lines 46 and 81 where it is written “atherosclerosis and so on”. What has been implied must be made explicit.
Minor points:
The use of the English language is not always correct. Some of the sentences are frankly incomprehensible. Moreover, several mistakes are present in the text. See for example lines 214 and 215 (what is the CAC?), line 297 (what is amoutr /?), and line 323 (TGF is the acronym for TRANSFORMING GROWTH FACTOR and not for TUMOR NECROSIS FACTOR).
Abbreviations are often misused. For example, the terms "ionizing radiation" or “reactive oxygen species” are sometimes abbreviated, while some other times they are not.
Reviewer 2 Report
The presented manuscript entitled 'Tissue Reactions and Mechanism in Cardiovascular Diseases Induced by Radiation' summarize the most recent literatures on the pathogenic pathways that contribute to the development of Radiation-induced cardiovascular diseases (RICVD) and provide preventative and therapeutic ideas.
1. All abbreviations in the Abstract should be disclosed.
2. Disseminated intravascular coagulation: It should be discussed whether disseminated intravascular coagulation development may be induced by ionizing radiation due to endothelial inflammation?
3. Line 233: ‘Japanese atomic bomb’: it is necessary to check the correct usage.
4. The section 5. ‘Prospective and conclusion’ is brief and should be extended. Therapies and methods for reducing vascular inflammation should be disclosed, including Fresh Frozen Plasma (FFP) transfusion previously used for therapy of radiation-affected patients.
5. It is necessary to recognize that the review is excellent and needs minimal corrections.
Round 2
Reviewer 1 Report
The manuscript has been revised in order to satisfy some of the criticisms raised against its original version. In its current version, the manuscript is undoubtedly more orderly and understandable, even if its reading is certainly not encouraged by having repeatedly addressed the same issues in the different paragraphs of the review.